# Spectral Norm Variance Regularization for Balanced Layer-Wise Dynamics in Training Deep Neural Networks

## Abstract

While deep neural networks (DNNs) are widely adopted in machine learning applications, addressing the systematic imbalance in their layer-wise training dynamics, manifested as discrepancies in learning behavior between earlier and deeper layers, remains a continuous research challenge for improving generalization ability. In this work, we propose spectral norm variance regularization (SNVR) to promote balanced layer-wise dynamics in training DNNs. The spectral norm of a hidden layer, corresponding to the largest singular value of its weight matrix, serves as a proxy for how well the layer has been trained. We introduce a regularization term that penalizes the variance of spectral norms across layers of the same type and incorporate it directly into the learning objective. SNVR requires no strong assumptions about network architecture or learning objective, and its use involves hyperparameter tuning only for controlling the regularization strength. Moreover, it does not significantly increase computational cost, since spectral norms can be efficiently approximated via the power iteration method. Experimental results on multiple benchmark datasets with various network architectures demonstrate that SNVR improves generalization performance compared with baseline methods. Further analyses of layer-wise spectral densities, gradient norms, and information plane provide additional evidence of its balancing effect. The source code is available at `https://anonymous.4open.science/r/spectral-norm-variance-regularization`.

## 1 Introduction

Deep Neural Networks (DNNs) have achieved remarkable success across diverse fields of machine learning, including computer vision, natural language processing, and speech recognition. This progress has largely been driven by the development of deeper and more complex architectures, which exhibit strong representational capacity and generalization ability. Alongside these advances, there has been growing research interest in understanding the layer-wise dynamics of training in DNNs (Tishby & Zaslavsky, 2015; Alain & Bengio, 2017; Cao et al., 2022; Dyballa et al., 2024). These studies highlight that training progresses unevenly across layers, revealing an inherent imbalance in how different layers learn. Chen et al. (2023) identified the layer convergence bias phenomenon, in which earlier layers converge faster than deeper layers. This asymmetry is aligned with the notion of spectral bias (Rahaman et al., 2019), which describes the tendency of neural networks to learn low-frequency components of the target function before gradually fitting higher-frequency components, even when the latter exhibit larger amplitudes.

The systematic imbalance in training dynamics across layers has important implications for both representation learning and generalization. While earlier layers readily fit low-frequency components, deeper layers struggle to fit the remaining higher-frequency components (Chen et al., 2023). This discrepancy can leave deeper layers insufficiently optimized, thereby causing the DNN to rely disproportionately on the representations learned by the early layers. Consequently, the overall representational capacity of the DNN is constrained, which can ultimately undermine its ability to generalize effectively. Empirical evidence supports this view. Rahaman et al. (2019) demonstrates that the DNN learns higher-frequency components later in training, accompanied by growth in the spectral norms of weight matrices. Chen et al. (2023) shows that deeper layers are associated with

sharper minima compared to earlier layers during training. Likewise, Lee (2024) finds that deeper layers tend to have larger gradient norms than earlier layers during training. These findings highlight the importance of balanced training dynamics across layers to stabilize optimization and promote robust generalization.

Many studies have attempted to improve the generation of DNNs by introducing mechanisms that use proxy indicators of layer-wise training quality to balance learning across layers. One representative line of work is layer-wise adaptive learning rates, in which the learning rate for each layer is dynamically adjusted according to layer-specific statistics such as gradient norms or variances (You et al., 2017; 2020; Ko et al., 2022; Zhou et al., 2023). Smaller learning rates are assigned to already stabilized layers, while larger learning rates are assigned to layers that remain un-converged and can benefit from more updates. Another line of work is layer-wise normalization, which explicitly normalizes weights or gradients across layers to encourage balanced training dynamics, either through hard constraints (Salimans & Kingma, 2016; Miyato et al., 2018) or by modifying learning objectives (Yoshida & Miyato, 2017; Xiao et al., 2023). Such normalization prevents certain layers from stagnating or dominating during training, thereby promoting more uniform convergence throughout the network. Although these methods have demonstrated promising empirical performance, they are largely heuristic in nature or depend on less direct control of training dynamics.

In this work, we propose Spectral Norm Variance Regularization (SNVR), a grounded way to achieve balanced layer-wise dynamics in training DNNs. The spectral norm of a hidden layer, corresponding to the largest singular value of its weight matrix, is known to increase as the layer learns to fit higher-frequency components of the target function (Rahaman et al., 2019). This spectral norm serves as a meaningful proxy for the scale of the spectral density of the layer, which reflects how well a layer has been trained (Mahoney & Martin, 2019; Martin & Mahoney, 2021; Martin et al., 2021; Yang et al., 2023). While spectral densities are computationally expensive to derive, spectral norms can be efficiently approximated (Yoshida & Miyato, 2017). Building on these insights, SNVR introduces a regularization term into the learning objective that penalizes variance in spectral norms across layers of the same type.

Our main contributions can be summarized as follows:

- We propose SNVR, which encourages all hidden layers to maintain similar spectral norms during training, leading to more balanced learning dynamics across layers.

- We empirically demonstrate that SNVR improves generalization performance compared with conventional methods across diverse benchmark datasets and network architectures.

- Through layer-wise analyses of spectral densities and gradient norms, we reveal imbalances in spectral norms and high variance in gradient norms during training, and demonstrate that SNVR effectively mitigates these imbalances.

- We show that SNVR enables deeper layers to capture more task-relevant information, supported by a theoretically grounded analysis based on Information Bottleneck (IB) principle (Tishby & Zaslavsky, 2015).

## 2 RELATED WORK

### 2.1 BALANCING LAYER-WISE DYNAMICS IN TRAINING DNNS

**Layer-Wise Adaptive Learning Rates** Several methods apply layer-specific learning rates instead of a single global rate to prevent earlier layers from converging too quickly. Most of these methods are specifically designed to improve stability and generalization in large-batch training of DNNs. LARS (You et al., 2017) dynamically scales the learning rate of each layer in proportion to the ratio between its weight norm and gradient update norm. LAMB (You et al., 2020) extends LARS to adaptive optimizers by incorporating layer-wise scaling with momentum. LENA (Ko et al., 2022) adjusts learning rate of each layer to be proportional to its gradient variance at each training iteration. For more general purposes, Temperature Balancing (TB) (Zhou et al., 2023) aligns the power-law (PL) exponents $\alpha$ of spectral densities across layers by assigning lower/higher learning rates to layers with smaller/larger $\alpha$, which correspond to more/less heavy-tailed spectral densities.

**Layer-Wise Normalization** These methods explicitly balance layer-wise properties to stabilize the training of DNNs, either by imposing hard constraints or by modifying the learning objective.

For methods based on hard constraints, normalization is enforced through post-processing of each hidden layer. Weight Normalization (WN) (Salimans & Kingma, 2016) reparameterizes individual weight vectors to decouple their norms from their directions. Spectral Normalization (SN) (Miyato et al., 2018) constrains the spectral norm of the weight matrix in each layer to 1, thereby bounding the Lipschitz constant and stabilizing the training of Generative Adversarial Networks.

For methods that modify the learning objective, differentiable regularization terms are introduced to guide layer-wise balance in training. Spectral Norm Regularization (SNR) (Yoshida & Miyato, 2017) employs the sum of the spectral norms across hidden layers as an additional regularization term. Xiao et al. (2023) propose two regularization terms: Weight Alpha Regularization (WAR) and Stable Rank Regularization (SRR) to encourage heavy-tailedness in the spectral densities of weight matrices. WAR minimizes the product of the PL exponent $\alpha$ and the log of the squared spectral norm across layers. SRR minimizes the stable rank, defined as the ratio of the sum of eigenvalues to the squared spectral norm of a spectral density, across layers.

## 2.2 REGULARIZATION ON SPECTRAL DENSITIES

For a hidden layer in a DNN, the empirical spectral density is defined as the distribution of eigenvalues of the Gram matrix of its weight matrix. These eigenvalues are identical to the squared singular values of the weight matrix. Heavy-Tailed Self-Regularization (HT-SR) theory (Mahoney & Martin, 2019; Martin & Mahoney, 2021) states that the weight matrix of each layer naturally evolves toward a heavy-tailed spectral density distribution as training progresses, reflecting an inherent form of self-regularization. The presence of heavy-tailed behavior in the spectral density can therefore serve as an empirical indicator of how well a layer has been trained (Martin et al., 2021; Yang et al., 2023). Grounded in this theory, many studies have explored regularization on spectral densities to achieve balanced training across layers.

Such regularization can be characterized by two aspects of spectral densities: scale and shape. Focusing on the scale, methods use spectral norms as indicators of the scale of spectral densities. For example, SN (Miyato et al., 2018) and SNR (Yoshida & Miyato, 2017) attempt to balance spectral norms by directly normalizing the weight matrix of each layer and by introducing a regularization term into the learning objective, respectively. Focusing on the shape, methods employ various indicators of spectral density and attempt to enforce balanced heavy-tailedness across layers. TB (Zhou et al., 2023) and WAR (Xiao et al., 2023) use an approximation of the PL exponent $\alpha$ of the spectral density, while SRR (Xiao et al., 2023) adopts the stable rank as a shape measure.

Regularizing the shape of spectral densities is a more direct way to encourage the spectral distributions of weight matrices to become heavy-tailed. However, deriving these shapes for certain weight matrices is computationally expensive (Zhou et al., 2023; Xiao et al., 2023). Moreover, shape indicators typically require multiple carefully tuned hyperparameters, which are often highly sensitive to the dataset and network architecture, making them difficult to apply in practice. The proposed method, SNVR, does not directly balance shape but instead minimizes the variance of spectral norms across layers of the same type. This is more practical, as spectral norms can be efficiently approximated at low cost and the regularization requires far fewer hyperparameters to be tuned.

## 3 PROPOSED METHOD

### 3.1 NOTATIONS

We denote a DNN, parameterized by a set of trainable parameters $\boldsymbol{\theta}$, by $f(\mathbf{X}; \boldsymbol{\theta})$. Let the DNN $f$ have $L$ parameterized hidden layers with weight tensors, excluding the output layer. For the set of all layer indices $\mathbb{H} = \{1, \ldots, L\}$, the weight tensor of the $\ell$-th hidden layer is represented as $\mathbf{W}^\ell$ for each $\ell \in \mathbb{H}$. The shape of $\mathbf{W}^\ell$ may vary depending on the layer type. If it is a linear layer, $\mathbf{W}^\ell$ is a 2-D tensor of shape $d_{\text{out}} \times d_{\text{in}}$, where $d_{\text{in}}$ and $d_{\text{out}}$ are the input and output dimensionalities, respectively. For a convolutional layer with a $k \times k$ kernel, $\mathbf{W}^\ell$ is a 4-D tensor of shape $c_{\text{out}} \times c_{\text{in}} \times k \times k$, where $c_{\text{in}}$ and $c_{\text{out}}$ are the numbers of input and output channels, respectively. We define subsets of layer indices corresponding to specific layer types as $\mathbb{G}_t \subset \mathbb{H}$. For example, in a ConvNeXt

architecture (Liu et al., 2022), which contains both convolutional and linear layers, its hidden layers are divided into two groups $\mathbb{G}_{\text{conv}}$ and $\mathbb{G}_{\text{linear}}$.

Given a training dataset as $D = \{(\mathbf{X}_i, \mathbf{y}_i)\}_{i=1}^n$, where $\mathbf{X}_i$ denotes the input and $\mathbf{y}_i$ is the corresponding label for the $i$-th instance, the learning objective is typically defined as:

$$\mathcal{J} = \frac{1}{n} \sum_{i=1}^n \mathcal{L}(f(\mathbf{X}_i), \mathbf{y}_i), \tag{1}$$

where $\mathcal{L}$ is a loss function. Common choices include cross-entropy for classification and squared error for regression.

## 3.2 Spectral Norms of DNNs

The spectral norm for a 2-D weight tensor $\mathbf{W}^\ell$ in a hidden layer, denoted by $\sigma(\mathbf{W}^\ell)$, is mathematically defined as:

$$\sigma(\mathbf{W}^\ell) = \max_{\|\mathbf{v}\|=1} \|\mathbf{W}^\ell \mathbf{v}\|, \tag{2}$$

which is equivalent to the largest singular value of $\mathbf{W}^\ell$. The calculation of the spectral norm for each layer depends on the shape of its weight tensor. If $\mathbf{W}^\ell$ has more than two dimensions, the spectral norm can be obtained by reshaping it into a 2-D tensor. For example, a 4-D weight tensor $\mathbf{W}^\ell$ in a convolutional layer is reshaped to $c_{\text{out}} \times c_{\text{in}} k^2$ to form a 2-D tensor.

For a fast approximation of $\sigma(\mathbf{W}^\ell)$, we employ the power iteration method as used in Yoshida & Miyato (2017). Starting with a random vector $\mathbf{v}$, it repeatedly applies the updates $\mathbf{u} \leftarrow \mathbf{W}\mathbf{v}$ and $\mathbf{v} \leftarrow \mathbf{W}^\top \mathbf{u}$. After a sufficient number of updates, the spectral norm can be approximated as:

$$\sigma(\mathbf{W}) \approx \|\mathbf{u}\|/\|\mathbf{v}\|. \tag{3}$$

The spectral norm of a hidden layer is an upper bound on its Lipschitz constant. Intuitively, it indicates how much some input direction gets stretched for the layer. For the entire DNN, the Lipschitz constant is bounded by the product of the spectral norms of all layers (Virmaux & Scaman, 2018). Thus, a DNN becomes less sensitive to input perturbations when the spectral norms of its layers are small (Yoshida & Miyato, 2017).

SNR (Yoshida & Miyato, 2017) builds on this insight by directly penalizing the growth of spectral norms during training. SNR does not impose hard constraints on the weight matrices but instead guides training through an additional term in the learning objective. The SNR term $\mathcal{L}_{\text{SNR}}$ is formulated as the sum of squared spectral norms across all hidden layers excluding the first:

$$\mathcal{L}_{\text{SNR}} = \frac{1}{2} \sum_{\ell \in \mathbb{H} \setminus \{1\}} \sigma(\mathbf{W}^\ell)^2. \tag{4}$$

By encouraging each layer to maintain a smaller spectral norm, this regularization effectively constrains the Lipschitz constant of the entire DNN, thereby improving its generalization and robustness against adversarial attacks.

## 3.3 Spectral Norm Variance Regularization (SNVR)

Owing to the systematic imbalance in training dynamics across layers, different degrees of heavy-tailed spectral density in each layer result in varying magnitudes and growth tendencies of spectral norms. While recent research (Zhou et al., 2023; Xiao et al., 2023) attempts to balance the shape of the spectral densities across hidden layers, deriving the spectral densities for all weight matrices is computationally very expensive and thus makes training substantially longer. Instead of directly treating the shape of spectral densities, we focus on balancing the spectral norms across hidden layers, from which we can benefit through their approximate computation and therefore avoid significantly increasing the training cost.

Motivated here, we propose SNVR, which introduces an explicit regularization term on the group-wise variance of spectral norms in the learning objective of the DNN. The SNVR term $\mathcal{L}_{\text{SNVR}}$ is

formulated as the group-wise squared deviation of each spectral norm $\sigma(\mathbf{W}^\ell)$ from the mean within its group $\mathbb{G}_t$:

$$\mathcal{L}_{\text{SNVR}} = \frac{1}{2} \sum_{\mathbb{G}_t \subset \mathbb{H} \setminus \{1\}} \sum_{\ell \in \mathbb{G}_t} (\sigma(\mathbf{W}^\ell) - \bar{\sigma}_{\mathbb{G}_t})^2 \tag{5}$$

where $\bar{\sigma}_{\mathbb{G}_t}$ is the mean spectral norm of weight matrices in the group $\mathbb{G}_t$. This regularization encourages the spectral norms of layers within the same group to maintain similar magnitudes throughout training.

For training a DNN, we use SNVR together with SNR as complementary regularization terms. We recommend applying these terms consistently with how the optimizer handles weight decay. When the optimizer does not decouple weight decay from the optimization with respect to the loss function (*i.e.*, standard L2 regularization is applied), the learning objective is explicitly modified to include both the SNR and SNVR terms:

$$\mathcal{J} = \frac{1}{n} \sum_{i=1}^{n} \mathcal{L}(f(\mathbf{X}_i), \mathbf{y}_i) + \frac{\lambda_{\text{WD}}}{2} \|\boldsymbol{\theta}\|^2 + \lambda_{\text{SNR}} \mathcal{L}_{\text{SNR}} + \lambda_{\text{SNVR}} \mathcal{L}_{\text{SNVR}}, \tag{6}$$

where $\lambda_{\text{WD}}$, $\lambda_{\text{SNR}}$ and $\lambda_{\text{SNVR}}$ are hyperparameters that control the strength of weight decay, SNR, and SNVR, respectively. For the optimizer with decoupled weight decay (*e.g.*, SGDW and AdamW) (Loshchilov & Hutter, 2017), the gradients of these regularization terms are directly incorporated into the parameter updates:

$$\boldsymbol{\theta} \leftarrow \boldsymbol{\theta} - \eta \Delta \boldsymbol{\theta} - \eta \nabla_{\boldsymbol{\theta}} \left( \frac{\lambda_{\text{WD}}}{2} \|\boldsymbol{\theta}\|^2 + \lambda_{\text{SNR}} \mathcal{L}_{\text{SNR}} + \lambda_{\text{SNVR}} \mathcal{L}_{\text{SNVR}} \right), \tag{7}$$

where $\Delta \boldsymbol{\theta}$ is the parameter update direction computed from the loss gradients using the optimizer and $\eta$ is the learning rate.

SNVR involves two hyperparameters in the learning objective, making it practically applicable while reducing the effort required for hyperparameter search. We provide a sensitivity analysis of the hyperparameter $\lambda_{\text{SNVR}}$ in Subsection 5.5.

## 4 EXPERIMENTS

### 4.1 DATASETS

We considered five widely used multi-class classification benchmark datasets with varying image resolutions. **CIFAR-10** and **CIFAR-100** consist of $32 \times 32$ images corresponding to 10 and 100 classes, respectively, with 50k images for training and 10k for testing. **TinyImageNet** comprises $64 \times 64$ images across 200 classes, with 100k images for training and 10k for testing. **STL-10** contains $96 \times 96$ images across 10 classes, with 5k images for training and 8k for testing. **Food-101** includes images of varying sizes across 101 classes, which we resized to $128 \times 128$, with 75k images for training and 25k for testing. For each benchmark dataset, all images were standardized to have zero mean and unit standard deviation per channel on the training set.

### 4.2 NETWORK ARCHITECTURES

We investigated the effectiveness on four network architectures with differing properties. We used **ResNet-18** and **ResNet-34** (He et al., 2016) with a minor modification. The first convolutional layer was changed to have a kernel size of $3 \times 3$ with stride 1, and the subsequent max pooling layer was removed. For both ResNet-18 and ResNet-34, all parameterized hidden layers were convolutional, consisting of 20 and 36 convolutional layers, respectively. Conv2NeXt-Tiny (**Conv2NeXt-T**) (Feng et al., 2022) is an adaptation of ConvNeXt (Liu et al., 2022) to low-resolution images. Its parameterized hidden layers consist of 32 convolutional layers and 24 linear layers.

### 4.3 IMPLEMENTATION DETAILS

**Data Augmentation** The data augmentation techniques used varied by network architecture. For ResNet-18 and ResNet-34, we applied random cropping and horizontal flipping. For ConvNeXt-T, we applied random cropping and horizontal flipping, RandAugment (Cubuk et al., 2020), Cut-Mix (Yun et al., 2019) with Beta$(0.5, 0.5)$, and Mixup (Zhang et al., 2018) with Beta$(0.5, 0.5)$.

**Training Configurations**   In the learning objective, we used cross-entropy as the loss function. The hyperparameters were set to $\lambda_{\text{SNR}} = 0.01$ and $\lambda_{\text{SNVR}} = 1.0$. For approximate computation of spectral norms, we set the number of updates to 5 in the power iteration method.

For ResNet-18 and ResNet-34, we used the SGD optimizer with an initial learning rate of 0.1, a momentum of 0.9, and a weight decay of $5 \times 10^{-4}$. Training was performed for 150 epochs using a cosine annealing learning rate scheduler. For Conv2NeXt-T, we used the AdamW optimizer with a weight decay of 0.05. The initial learning rates for Conv2NeXt-T was $4 \times 10^{-3}$. Training was performed for 150 epochs using a cosine annealing learning rate scheduler, with a linear warm-up over the first 5 epochs. Label smoothing of 0.1 was applied during training. For both optimizers, the minibatch size varied by dataset. The minibatch size was set to 256 for CIFAR-10 and CIFAR-100, 128 for TinyImageNet, 64 for STL-10 and Food-101.

**Evaluation Protocol**   The performance was evaluated in terms of Top-1 accuracy. All experiments were repeated five times with different random seeds, and the average and standard deviation of the results were reported. We used an NVIDIA A100 GPU with 40GB of memory for the Food-101 dataset and an NVIDIA RTX3090 GPU with 24GB of memory for the others.

### 4.4   BASELINE METHODS

We compared **SNVR** with 7 baseline methods introduced in Subsection 2.1. We considered **WD** as the most basic and simplest form of regularization. For methods that perform layer-wise normalization through hard constraints, we included **WN** (Salimans & Kingma, 2016) and **SN** (Miyato et al., 2018). For layer-wise normalization by modifying the learning objective, we included **SNR** (Yoshida & Miyato, 2017), **SRR** (Xiao et al., 2023), and **WAR** (Xiao et al., 2023). For layer-wise adaptive learning rates, **TB** (Zhou et al., 2023) was adopted. It should be noted that **WD** and **SNR** (Yoshida & Miyato, 2017) can be regarded as ablations of **SNVR**.

The configurations of each baseline method were set identical to the defaults in their original implementations, unless otherwise specified. For SNR, SRR, and WAR, the regularization strength hyperparameters $\lambda_{\text{SNR}}$, $\lambda_{\text{SRR}}$, and $\lambda_{\text{WAR}}$ were set to 0.01, 0.001, and 0.00001, respectively. These values were selected from the search space $\{0.1, 0.01, 0.001, 0.0001, 0.00001\}$ to optimize the performance of ResNet-34 on CIFAR-100.

## 5   RESULTS AND DISCUSSION

### 5.1   GENERALIZATION PERFORMANCE

Table 1 compares the generalization performance of the baseline and proposed methods in terms of Top-1 accuracy. The overall results demonstrate the effectiveness of SNVR relative to the baseline methods, with SNVR achieving the highest average rank across datasets for all network architectures. In particular, SNVR consistently outperforms its ablation SNR as well as baseline methods that regulate the shapes of spectral densities of layers, including TB, WAR, and SRR. Within the ResNet family, the effectiveness of SNVR is more pronounced on the deeper ResNet-34 compared to ResNet-18, where handling layer-wise imbalance would have a stronger effect.

### 5.2   LAYER-WISE SPECTRAL DENSITIES

We investigated the layer-wise spectral densities of DNNs as an indicator of balance across layers. Figure 1 visualizes the singular value distribution and spectral norm at each layer of Conv2NeXt-T trained on CIFAR-100 using different methods. For each method, we report the slope of spectral norms with respect to layer indices for each layer type. Among the methods, WD shows a tendency for spectral norms to increase in deeper layers, while most other methods yield reduced slopes by balancing spectral norms across layers. While Conv2NeXt-T is composed of both convolutional and linear layers, the magnitudes of spectral norms differ between the two types. Linear layers exhibit higher values than convolutional layers, highlighting the need for balancing within individual groups. SNVR addresses this by directly minimizing the variance of spectral norms across layers within each group, achieving slopes close to zero for both layer types. Similarly, SN yields exact zero slopes because it explicitly normalizes the spectral norm to 1 in every layer. In contrast, TB,

Table 1: Generalization performance comparison on different datasets and network architectures. Each table entry represents "mean±std. dev." of Top-1 accuracy. **Bold** indicates the best performance, and underline indicates the second-best.

| | Method | CIFAR-10 | CIFAR-100 | TinyImageNet | STL-10 | Food-101 | Avg. Rank |
|---|---|---|---|---|---|---|---|
| **ResNet-18** | **WD** | **96.28±0.15** | 80.11±0.21 | 68.68±0.23 | 87.93±0.43 | 85.82±0.09 | 3.2 |
| | **WN** (2016) | 96.22±0.12 | 80.02±0.23 | 68.45±0.17 | 87.53±0.36 | 77.58±4.39 | 5.8 |
| | **SN** (2018) | 96.09±0.11 | 79.96±0.14 | 69.03±0.36 | 87.71±0.54 | 85.88±0.10 | 4.0 |
| | **TB** (2023) | 96.01±0.20 | 79.50±0.26 | 67.98±0.27 | 86.04±0.48 | 85.68±0.08 | 7.2 |
| | **WAR** (2023) | 96.13±0.08 | 80.15±0.21 | 68.73±0.29 | 87.63±0.40 | 85.76±0.01 | 3.8 |
| | **SRR** (2023) | 95.90±0.06 | 79.84±0.19 | 68.45±0.08 | 86.50±0.96 | 85.60±0.13 | 6.8 |
| | **SNR** (2017) | 96.27±0.15 | 80.03±0.20 | 68.74±0.23 | **90.01±0.23** | 85.56±0.09 | 3.4 |
| | **SNVR** (Ours) | 96.25±0.02 | **80.25±0.26** | **69.15±0.33** | 89.75±0.53 | **85.94±0.07** | **1.6** |
| **ResNet-34** | **WD** | 96.59±0.08 | 81.71±0.21 | 71.94±0.21 | 86.35±1.16 | 87.55±0.25 | 4.0 |
| | **WN** (2016) | 96.46±0.08 | 81.31±0.04 | 71.63±0.35 | 85.35±2.25 | 80.75±5.15 | 7.2 |
| | **SN** (2018) | 96.51±0.11 | 81.46±0.10 | 72.22±0.16 | 85.84±1.22 | 87.62±0.39 | 4.4 |
| | **TB** (2023) | 96.32±0.07 | 81.38±0.17 | 71.22±0.11 | 83.80±1.33 | 87.66±0.09 | 6.6 |
| | **WAR** (2023) | 96.62±0.12 | 81.74±0.08 | 71.92±0.57 | 86.27±0.47 | 87.47±0.20 | 4.0 |
| | **SRR** (2023) | 96.38±0.11 | 81.36±0.30 | 71.92±0.27 | 85.65±0.24 | 87.32±0.27 | 6.4 |
| | **SNR** (2017) | 96.62±0.04 | 81.80±0.10 | 72.58±0.11 | 90.39±0.21 | 87.90±0.03 | 2.0 |
| | **SNVR** (Ours) | **96.65±0.06** | **82.09±0.04** | **72.62±0.16** | **90.70±0.14** | **88.18±0.07** | **1.0** |
| **Conv2NeXt-T** | **WD** | 97.36±0.12 | **83.23±0.26** | 71.83±0.12 | 89.66±0.52 | - | 3.0 |
| | **WN** (2016) | 96.74±0.16 | 81.68±0.45 | 69.33±0.18 | 85.61±1.02 | - | 7.3 |
| | **SN** (2018) | 96.53±0.03 | 81.08±0.48 | 70.50±0.17 | 81.17±2.25 | - | 7.8 |
| | **TB** (2023) | 97.33±0.08 | 83.22±0.08 | 71.51±0.10 | 89.68±0.38 | - | 3.3 |
| | **WAR** (2023) | 97.20±0.05 | 82.95±0.49 | 70.73±0.41 | 89.50±0.24 | - | 6.0 |
| | **SRR** (2023) | 97.28±0.14 | 83.05±0.53 | 71.28±0.47 | 89.67±0.14 | - | 4.8 |
| | **SNR** (2017) | 97.44±0.06 | 83.21±0.29 | 72.06±0.02 | 90.29±0.31 | - | 2.3 |
| | **SNVR** (Ours) | **97.46±0.14** | 83.19±0.43 | **72.07±0.17** | **90.40±0.30** | - | **1.8** |

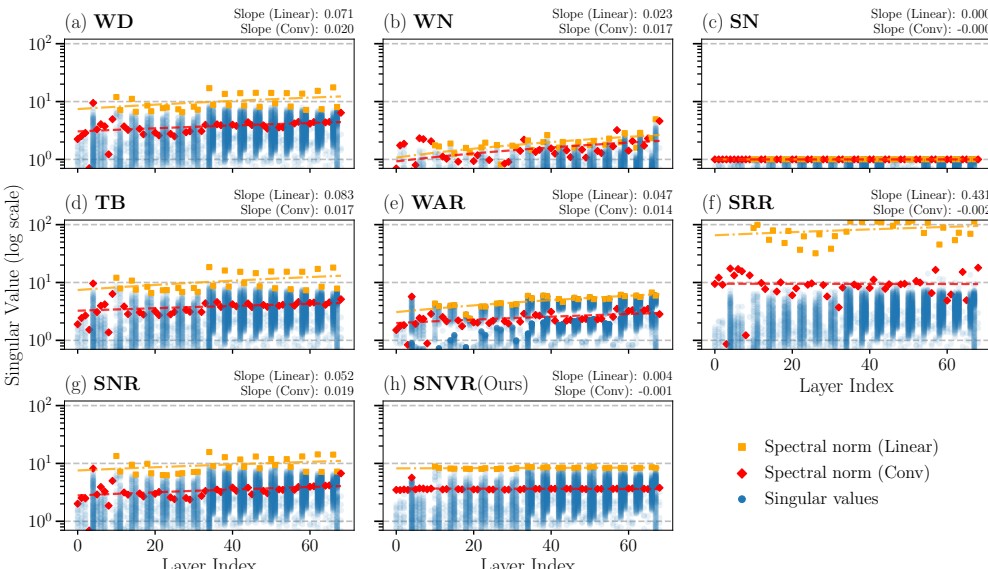

Figure 1: Visualization of layer-wise singular value distributions for Conv2NeXt-T trained on CIFAR-100. Spectral norms are color-coded by layer type. The slope for each layer type is obtained by fitting a linear function to the log spectral norms against layer indices.

which attempts to balance the layer-wise shapes of spectral densities, produces even larger slopes than WD.

## 5.3 LAYER-WISE GRADIENT NORMS

We also investigated the layer-wise gradient norms of DNNs. Figure 2 visualizes the gradient norms across layers for ResNet-34 trained on CIFAR-100 over 150 epochs. There is an overall tendency

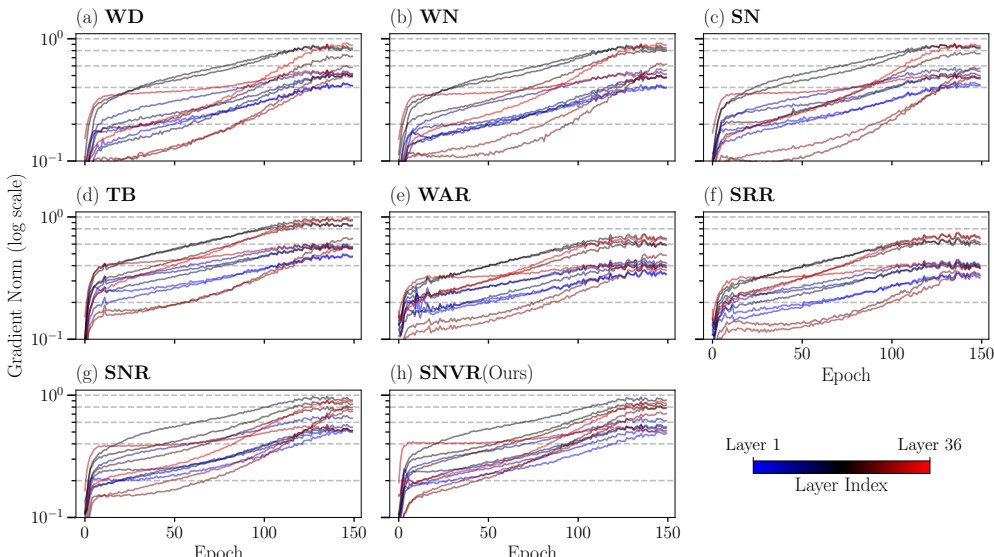

Figure 2: Visualization of layer-wise gradient norms for a ResNet-34 model trained on CIFAR-100 dataset. Each curve represents the gradient norm of a specific layer, averaged over the training data for each epoch. The layers are sampled at an interval of three and color-coded according to their depth.

for high variance in gradient norms across layers, especially in the later stages of training. The methods regularizing spectral densities, including TB, SNR, and SNVR, helped reduce this variance during training. For SNVR, the spread of gradient norms across layers narrowed more consistently as training progressed compared with the baselines. This suggests that SNVR stabilizes the dynamics of gradient flow while effectively reducing inter-layer discrepancies.

## 5.4 INFORMATION PLANE ANALYSIS

The information plane (Shwartz-Ziv & Tishby, 2017) visualizes how a DNN learns through the trade-off between compression and prediction, grounded in IB principle (Tishby & Zaslavsky, 2015). Given input $X$ and output $Y$, it plots the mutual information $I(X;T)$ against $I(T;Y)$ for each hidden layer $T$ as training progresses. $I(X;T)$ represents how much information the layer $T$ preserves from the input $X$ (*i.e.*, compression), while $I(T;Y)$ indicates how much information the layer $T$ conveys about the output $Y$ (*i.e.*, prediction). Following the experimental setup of Shwartz-Ziv & Tishby (2017), we trained a simple feedforward neural network, comprising 6 fully-connected hidden layers with 10-7-5-4-3-2 units and tanh activation, on a synthetic dataset with 12 input features for binary classification.

Figure 3 shows the information planes obtained with WD, SNR, and SNVR. SNVR yielded balanced $I(T;Y)$ values across hidden layers, gradually moving toward higher $I(T;Y)$ as training progressed. By the end, all layers converged to similarly high $I(T;Y)$, while deeper layers approached smaller $I(X;T)$. This behavior reflects consistent predictive information across layers with stronger compression in deeper layers, which highlights the advantage of SNVR in utilizing depth effectively in line with the IB principle. In contrast, WD and SNR exhibited uneven distributions across layers. Earlier layers had a rapid increase in $I(T;Y)$ at the initial training stage, while deeper layers remained at relatively low $I(T;Y)$ and high $I(X;T)$ by the end of training, indicating less effective utilization of depth.

## 5.5 HYPERPARAMETER SENSITIVITY ANALYSIS

For SNVR, we analyzed the sensitivity of the hyperparameter $\lambda_{\text{SNVR}}$ with respect to generalization performance. Table 2 shows the results for varying values of $\lambda_{\text{SNVR}} \in \{0, 0.01, 0.1, 1, 10\}$. Overall, SNVR maintained stable performance around the default hyperparameter setting, suggesting that it

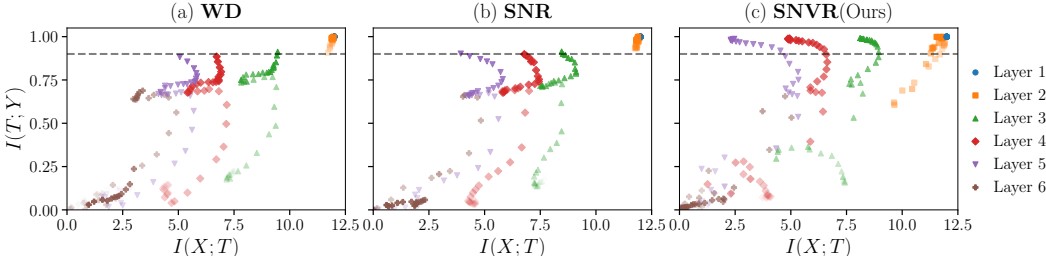

Figure 3: Visualization of information planes, following Shwartz-Ziv & Tishby (2017)'s work. Point color and shape distinguish hidden layers. Point transparency represents training progress, with lighter/darker shades indicating earlier/later epochs.

Table 2: Sensitivity analysis of the hyperparameter $\lambda_{\text{SNVR}}$ on SNVR. The default setting is marked with *. Each table entry represents "mean±std. dev." of Top-1 accuracy. **Bold** indicates the best performance, and underline indicates the second-best.

| | **ResNet-18** | | | **ResNet-34** | | | **Conv2NeXt-T** | | |
|---|---|---|---|---|---|---|---|---|---|
| $\lambda_{\text{SNVR}}$ | CIFAR-10 | CIFAR-100 | STL-10 | CIFAR-10 | CIFAR-100 | STL-10 | CIFAR-10 | CIFAR-100 | STL-10 |
| 0 | 96.27±0.15 | 80.03±0.20 | 89.86±0.22 | 96.62±0.04 | 81.80±0.10 | 90.39±0.21 | 97.44±0.06 | 83.21±0.29 | 90.29±0.31 |
| 0.01 | **96.41±0.03** | 79.98±0.13 | 89.70±0.21 | 96.62±0.04 | **82.15±0.26** | 90.52±0.22 | 97.37±0.05 | 83.46±0.43 | 90.39±0.20 |
| 0.1 | 96.33±0.23 | 80.24±0.12 | **90.02±0.14** | **96.70±0.05** | 81.97±0.11 | 90.62±0.20 | **97.49±0.11** | 83.41±0.31 | 90.49±0.19 |
| * 1 | 96.25±0.02 | **80.25±0.26** | 89.92±0.28 | 96.65±0.06 | 82.09±0.04 | **90.70±0.14** | 97.46±0.14 | 83.19±0.43 | 90.40±0.30 |
| 10 | 96.04±0.11 | 79.17±0.16 | 88.22±0.39 | 96.37±0.05 | 81.37±0.08 | 87.80±0.15 | 97.40±0.09 | **83.52±0.12** | **90.50±0.32** |

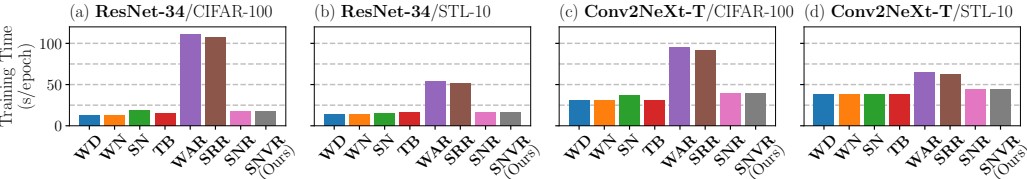

Figure 4: Per-epoch training time comparison across methods for ResNet-34 and ConvNeXt-T on CIFAR-100 and STL-10.

can be applied without careful task-specific hyperparameter tuning. SNVR consistently improved over SNR (*i.e.*, when $\lambda_{\text{SNVR}} = 0$) in most cases.

## 5.6 TRAINING TIME

Figure 4 compares per-epoch training times across methods for training ResNet-34 and Conv2NeXt-T on CIFAR-100. SNVR does not incur a significant increase in training time. In contrast, methods that compute shape indicators of spectral densities at each training iteration, such as WAR and SRR, require significantly longer training times.

## 6 CONCLUSION

We proposed SNVR, a simple yet effective regularization method for balanced layer-wise dynamics in training DNNs. SNVR explicitly enforced layer-wise balance by incorporating a regularization term into the learning objective, minimizing the variance of spectral norms across layers of each type. Experimental results on various datasets and network architectures demonstrated that SNVR achieved comparable or better generalization performance than baseline methods. Furthermore, analysis of layer-wise spectral densities and information planes qualitatively confirmed the balancing effect.

## ETHICS STATEMENT

Our method is broadly applicable to a wide range of architectures. It could potentially be misused to optimize models for malicious purposes, such as stabilizing the training of models intended for malicious or unethical applications. We explicitly deny any such use of this research and are committed to its responsible application.

## REPRODUCIBILITY STATEMENT

Our work is publicly available on https://anonymous.4open.science/r/spectral-norm-variance-regularization. To aid reproducibility, we have included the complete source code along with all configuration files used in our experiments. The analysis results presented in this paper can also be fully reproduced using the provided scripts.

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

---

**Algorithm 1** Spectral norm variance regularization

---

1: **Initialize:** model parameters $\theta$.
2: **for** each training iteration **do**
3:     Sample minibatch $\{(\mathbf{X}_i, \mathbf{y}_i)\}_{i=1}^n$.
4:     $\nabla_\theta \mathcal{L}_{\text{task}} \leftarrow \nabla_\theta \frac{1}{n} \sum_{i=1}^n \mathcal{L}(f(\mathbf{X}_i), \mathbf{y}_i)$.
                                                        $\triangleright$ 1. Spectral norm approximation
5:     **for** each layer $\ell \in \mathbb{H} \setminus \{1\}$ **do**
6:         Initialize random vector $\mathbf{v}^\ell$.
7:         **for** $p = 1, \ldots, P$ **do**
8:             $\mathbf{u}^\ell \leftarrow \mathbf{W}^\ell \mathbf{v}^\ell$
9:             $\mathbf{v}^\ell \leftarrow (\mathbf{W}^\ell)^\top \mathbf{u}^\ell / \|(\mathbf{W}^\ell)^\top \mathbf{u}^\ell\|$
10:         **end for**
11:         $\sigma(W^\ell) \leftarrow \|\mathbf{u}^\ell\| / \|\mathbf{v}^\ell\|$
12:     **end for**
                                                         $\triangleright$ 2. Group-wise averages
13:     **for** each group $\mathbb{G}_t \subset \mathbb{H}$ **do**
14:         $\bar{\sigma}_{\mathbb{G}_t} \leftarrow \frac{1}{|\mathbb{G}_t|} \sum_{\ell \in \mathbb{G}_t} \sigma(W^\ell)$
15:     **end for**
                                                       $\triangleright$ 3. Regularization gradients
16:     **for** each $\ell \in \mathbb{H} \setminus \{1\}$ **do**
17:         Find $t$ s.t. $\ell \in \mathbb{G}_t$.
18:         $\nabla_{\theta^\ell} \mathcal{L}_{\text{SNR}} \leftarrow \sigma(W^\ell) \mathbf{u}^\ell (\mathbf{v}^\ell)^\top$
19:         $\nabla_{\theta^\ell} \mathcal{L}_{\text{SNVR}} \leftarrow \left(\sigma(W^\ell) - \bar{\sigma}_{\mathbb{G}_t}\right) \mathbf{u}^\ell (\mathbf{v}^\ell)^\top$
20:     **end for**
                                                           $\triangleright$ 4. Parameters update
21:     **if** not `decouple` **then**
22:         $\nabla_\theta \mathcal{L}_{\text{total}} \leftarrow \nabla_\theta \mathcal{L}_{\text{task}} + \lambda_{\text{WD}} \theta + \lambda_{\text{SNR}} \nabla_\theta \mathcal{L}_{\text{SNR}} + \lambda_{\text{SNVR}} \nabla_\theta \mathcal{L}_{\text{SNVR}}$.
23:         $\Delta\theta \leftarrow \texttt{Momentum}(\nabla_\theta \mathcal{L}_{\text{total}})$.
24:     **else**
25:         $\Delta\theta \leftarrow \texttt{Momentum}(\nabla_\theta \mathcal{L}_{\text{task}})$.
26:         $\Delta\theta \leftarrow \Delta\theta + \lambda_{\text{WD}} \theta + \lambda_{\text{SNR}} \nabla_\theta \mathcal{L}_{\text{SNR}} + \lambda_{\text{SNVR}} \nabla_\theta \mathcal{L}_{\text{SNVR}}$.
27:     **end if**
28:     Update: $\theta \leftarrow \theta - \eta \Delta\theta$.
29: **end for**

---

# A   ALGORITHM

Algorithm 1 describes the procedure of adapting SNVR to a conventional optimization process. The operator $\texttt{Momentum}(\cdot)$ depends on the optimizer, such as SGD or Adam. The flag `decouple` indicates whether the regularization gradients are included in the momentum calculation.

# B   LAYER-WISE SPECTRAL DENSITIES

Figure 5 visualizes the singular value distribution and spectral norm at each layer of ResNet-34 trained on CIFAR-100. Our proposed method, SNVR, effectively mitigates some spectral norms being prominent, achieving a uniform distribution of spectral norms across all layers, as indicated by a slope close to zero.

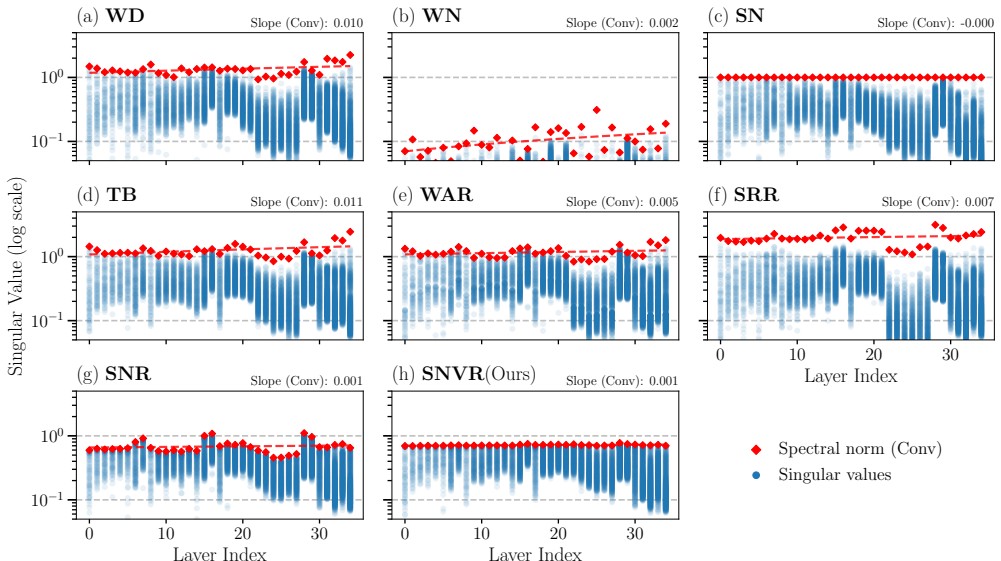

Figure 5: Visualization of layer-wise singular value distributions for ResNet-34 trained on CIFAR-100. The slope for each layer type is obtained by fitting a linear function to the log spectral norms against layer indices.

