# OpenReview forum: "Spectral Norm Variance Regularization for Balanced Layer-Wise Dynamics in Training Deep Neural Networks"
_ICLR.cc/2026/Conference — ICLR 2026 Conference Withdrawn Submission_

### Official Review · Reviewer_F9Fn · 2025-10-27

**Soundness:** 2
**Presentation:** 3
**Contribution:** 3
**Rating:** 4
**Confidence:** 2

**Summary:**

The paper proposes a new regularization/normalization technique that aims at combating uneven training across (nested) layers: The idea is to estimate the spectral norm of each weight matrix (implemented conveniently using a few cycles of power-iteration) and penalize the spectral norm, i.e., the magnitude of the largest eigenvalue, in order to obtain a more equal training. This could also be seen as an implicit learning rate scheduler.
The method works well in practice (tested on common, smaller image recognition benchmarks). The margin to other methods does not seem to be big, but there seems to be a consistent performance improvement, and one would not expect huge jumps for these types of techniques. The paper also provides some analysis and potential explanation; I personally would see some issues here (see below).

**Strengths:**

The paper is well written and the method is easy to understand. The idea is well motivated, although I think that the rather straightforward motivation is missing a few subtle but crucial aspects. I do think that it is plausible that this method would indeed help in getting more performance out of the training process, but not for the reason the paper suggests (I will explain this in the next section). The results are good. Although one could always ask for more (data sets, architectures, experiments), I am willing to believe that one sees a small improvement when used as a drop-in add-on, which is nonetrivial to obtain at the current state of the art. This aspect (a new method that actually seems to help, and is not terribly difficult to implement and use) seems to be the main strength of the paper in my opinion, and one could consider acceptance.

**Weaknesses:**

My main issue with the paper is the explanation for the effect: The authors claim to address the uneveness of training across multiple network layers, and I would agree with this assessment in my understanding of the situation. However, the effects of uneven training, which I would broadly frame as "gradient magnitude explosions", depend decisively on the architecture employed: First of all, tests are performed with networks with shortcut connections (ResNet and descendants), which dampen these effects through the shortcuts: Here, deeper paths are down-weighted so that the issues with nesting deep hierarchies of layers are rather mild, empirically and theoretically. As a side note at this point: I think that it would be very insightful to repeat the benchmarks with a non-residual network such as VGG-19 (with and without BN), or a ResNet with shortcuts turned off, which greatly accentuates the effect and shows more clearly whether training speed disparities are kept under control.
Second, and more importantly, the networks considered employ batch-normalization (or layer-norm), which has the known and unintuitive effect of counteracting the balancing of the gradient magnitude achieved by the initialization (such as He-init for ReLU, or its analogon for other nonlinearities). As shown in [Yang et al., ICLR 2019] ("A Mean Field Theory Of Batch Normalization"), the magnitude of gradients explodes exponentially towards the input layers (this can be understood more easily/intuitively by just repeating the He-deriviation with the shifts of the BN-statistics, as shown for example in a blog post by K. Luther, 2020). At the same time, batch normalization induces a scale invariance on the weights, i.e., scaling the whole matrix (and its top eigenvalue) up does not change its effect; in contrast, under fixed learning steps, the effective learning speed actually becomes lower. This effect has been studied by several recent papers (I am aware of recent one from Göpel et al. at ICML 2024 that also cites earlier work on this matter); in particular, this leads to a jo-jo effect where the bottom layers first overshoot in magnitude and then effectively freeze in the next step. This would not be visible by looking only at the layer norms without taking gradient magnitudes into account. It has also been shown that this negative feedback loop leads to an eventual equalization, which can be reached in practice by the LARS-strategy, weight normalization, or small (well-controlled) step-sizes, all of which being (again, unintuitively) essentially equivalent due to the weight-scale symmetry induced by BN. Without BN, exploding gradients can also happen, but in my understanding these are artifacts of disparities in in- and out-degrees of linear layers which makes it impossible to choose a good intialization. (There is more to this; see for example "deep kernel shaping" and the classic "dynamical isometry" papers [Xiao et al. ICML 2018].)

What I would guess is happening (I am not very sure though, as this is a big area full of subtle effects, and my personal knowledge and experience is limited to a small corner of it) is the following: While the disparities in effective learning steps can be mitigated with existing techniques such as LARS or the similar (weight decay can also play a role here, as it can "unfreeze" layers, when parametrized well, even without normalization), there is also the effect of a successive loss of dimensionality (stacks of linear layers lead to a concentration of the overall singular value spectrum of their concatenation, and this is hard to avoid, again see "dynamical isometry"). It seems plausible to me (although I did not do a careful analysis at this point), that penalizing the top singular value would have an (additional) equalizing effect on the spectrum of the whole stack of layers, unlike the techniques that just adjust layer-wise weightnorms/learning rates. That in turn could be responsible for an improvement in performance.

I would like to emphasize that the last part is mostly speculation; my criticism is that the effect of normalization (BN in ResNet and LN in ConvNext) and short-cut connections is not explored in the analysis part, and that one could probably examine and explain more deeply what is going on. I am not very convinced by the information bottleneck argument – is there some reference showing that this is indeed a crucial effect? My understanding has been that this has been rather hypothetical so far, but I might have easily missed something important here (in that case, it would be good to emphasize this in the text).

In summary, I my impression is that this is a method with "potentially some potential", which is nontrivial for these matters, but the paper would benefit from a bit more work on the analysis side. Note: My confidence that I understood everything correctly is moderate to low, as there are many subtleties and I only know a small excerpt of the literature.

**Questions:**

Did you try a deep network without shortcuts / residual connections? If your idea is correct, the improvement should be much more dramatic here (and if so, that would strengthen the paper a lot, even if this is rare in practice).
Did you try out networks with and without normalization layers? More generally, are the effects of normalization important for the (understanding of the) proposed method?
Did the method haven an influence of the overall spectral behavior of the network (expected variability SVD of the Jacobian of the overall network at training points)?

---

### Official Review · Reviewer_nxpH · 2025-10-31

**Soundness:** 1
**Presentation:** 4
**Contribution:** 2
**Rating:** 2
**Confidence:** 3

**Summary:**

The authors propose a novel regularization method based on penalizing the variance of the spectral norm across hidden layers in neural networks, to alleviate pathological issues in training deep layers in neural networks. The authors explain how to efficiently compute the loss term and evaluate it over fully-connected and convolutional layers. The method is evaluated on the CIFAR-10, CIFAR-100, TinyImageNet, STL-10 and Food-101 datasets, with models including ResNet-18, 34 and Conv2NeXt-T.

**Strengths:**

* The paper is very well written, and in a very accessible manner.
* The background is especially well written, with some excellent references, and was really a joy to read.
* The method outlined is intuitive
* The authors address the issue of computational cost well
* The authors do explore the hyper parameters (two main ones associated with the SNR and SNVR loss terms)

**Weaknesses:**

While the methodology is very well motivated and explained, the empirical analysis and results are overall unfortunately poor and unconvincing, and there are obvious missing ablations:
* The results are demonstrated only on relatively small and shallow image classification models --- this is especially surprising given the motivation of the work on addressing issues with learning deep layers. I will note that even using an extremely deep ResNet model for CIFAR-10 would be more convincing.
* Despite the authors claims, in Table 1, it's clear that the improvements in generalization for the deeper/larger models are actually within variance, as compared to the shallower models (e.g. ResNet 18). This is the opposite of what would be expected given the motivation of the method.
* Given the computational resources the authors used for the experiments (i.e. a 40GB A100), it's not obvious why results on even modest models/datasets (ResNet-50/ImageNet) were used, never mind much deeper and larger Vision Transformer models.
* There is no ablation shown explaining for example why the SNR loss term is needed, and what the relative impact of the presence of the SNR vs. SNVR loss terms are (aside from the loss hyperparameter sweep). This is especially relevant as the weighting for the SNR term is 0.01 vs. 1.0 for SNVR, a stark difference.
* The ResNet models used are modified ImageNet models, not CIFAR-10/100 models. Not clear why the authors did not use the ResNet models actually designed for CIFAR-10 from the ResNet paper (i.e. ResNet-20,-32,-44,-56,-110 or 1202)
* The authors optimize the hyper parameters for their method (SNVR), while using default hyper parameters optimized in a different data/model setting for the baselines, clearly putting the baselines at a disadvantage.
* Having four different loss terms can be problematic for hyper parameter optimization, although I believe one of these (WD) is just standard weight decay unless I'm misunderstanding it.

Minor feedback:
* When discussing "groups" in the context of filter groups in ConvNeXt models, perhaps use the term "filter groups" instead as it will not be obvious to most people what this means/that it's not the more general usage of the term.
* Might be better to explicitly define or use $\mathbf{v}$ in equation 2 before it's used, unless I've missed the definition. It becomes more clear in context later though.

**Questions:**

* Why is the SNR term required? Are there any ablation results demonstrating the relative impact of having the SNR term or not is (and for that matter the SNVR term).
* Why are results on deeper models not evaluated? Are there computational constraints in the methodology not explained in the paper/that I missed that makes these infeasible to run?
* Would the results be better in models not using skip connections or normalization (i.e. batch norm) as all the models used do use both I believe?

---

### Official Review · Reviewer_wGMP · 2025-11-01

**Soundness:** 3
**Presentation:** 3
**Contribution:** 2
**Rating:** 4
**Confidence:** 4

**Summary:**

This paper proposes Spectral Norm Variance Regularisation (SNVR), a method that introduces a regularisation term which penalises the variance of spectral norms among hidden layers that are of the same type. The method aims to encourage the spectral norms of all similar layers to maintain a consistent magnitude throughout the training process, thereby promoting more balanced learning dynamics across layers.

**Strengths:**

1. Intuitive Idea: The core idea of using variance as a metric for balance is highly intuitive.
2. Low Computational Cost: The implementation complexity is low, as the spectral norms can be approximated efficiently using the power iteration method.
3. Broad Applicability: The method does not rely on strong assumptions about the network architecture or the learning objective, suggesting it could be applied widely.

**Weaknesses:**

1. Justification of the Core Assumption: The central hypothesis, that all layers of the same type should have similar spectral norms, is not sufficiently explored or justified. The paper relies more on empirical validation than on a discussion of the theoretical grounding or potential limitations of this assumption.
2. Generalisability of SNVR: The paper's experiments are confined to a limited scope, focusing on small to medium-scale vision datasets like CIFAR-10/100 and STL-10. The models used, such as ResNet-18/34 and Conv2NeXt-T, are also relatively small convolutional networks. A key limitation is the absence of validation on more demanding benchmarks. The effectiveness of SNVR has not been tested on: Large-scale datasets, such as ImageNet-1k or Non-convolutional architectures, such as Vision Transformers (ViTs) or Large Language Models (LLMs). This narrow experimental setup makes it difficult to assess the scalability and general applicability of SNVR.
3. Missing Setting: In the paper, SNVR is always used in combination with SNR. This leaves a significant gap, as there is no study to demonstrate the standalone effect of SNVR. If SNVR must be used with SNR, the paper lacks a deep explanation for why this dependency exists.
4. Grouping Strategy: The strategy for grouping layers is fixed by layer type. This choice is not deeply justified over other potential strategies, and the paper does not explore the method's sensitivity to how layers are grouped.

**Questions:**

1. Could you provide a more in depth theoretical argument as to why the spectral norms of all layers of the same type should be uniform? Are there any architectures or tasks where the optimal state might involve a systematic variation in spectral norms? In such cases, could forcing uniformity via SNVR potentially restrict the model's performance?
2. Do you have any empirical results to suggest that SNVR is applicable and effective for large scale datasets, models and other architectures, such as Transformers?
3. Is it essential for SNVR to be used with SNR? Could you provide ablation experiments that isolate the contribution of SNVR? If the combination is necessary, what is the theoretical or empirical reasoning for this?
4. How sensitive is the SNVR method to the choice of grouping strategy? Could this grouping be made data driven or even learnable?

---

### Official Review · Reviewer_feNq · 2025-11-01

**Soundness:** 2
**Presentation:** 2
**Contribution:** 1
**Rating:** 2
**Confidence:** 3

**Summary:**

This paper proposes Spectral Norm Variance Regularization (SNVR), which adds a regularization term penalizing the variance of spectral norms across layers of the same type to achieve more balanced layer-wise training dynamics. The authors claim that SNVR encourages uniform learning progress across layers, improves generalization, and does not increase computational cost. Experiments on several small-scale vision benchmarks (CIFAR, TinyImageNet, STL, Food-101) with ResNet and Conv2NeXt architectures are presented, along with some analyses on spectral densities, gradient norms, and information planes.

**Strengths:**

The charts are beautifully presented, the paper is formatted correctly, and the color scheme is pleasing.

The proposed regularization term is simple to implement and easily integrates with existing training pipelines.

The overall presentation is clear, and the method is computationally lightweight.

**Weaknesses:**

The derivation from Eq. (6) to Eq. (7) is not rigorous. The paper does not show how the gradients of the regularization terms are propagated into the parameter updates — it simply inserts them into the update rule without mathematical justification.

The coefficients $\lambda_{WD}$, $\lambda_{SNR}$, and $\lambda_{SNVR}$ appear in both the coupled and decoupled optimization settings, but their physical meaning and scaling relationship in these two contexts are never clarified.

The concept of Layer-Wise balance has been extensively explored in optimizers, OOD generalization, catastrophic forgetting, and feature design. Thus, the claimed novelty here appears limited.

The experimental networks are too simple. Using only ResNet and Conv2NeXt is far from sufficient to support the claimed general applicability.

The compared methods are quite outdated — the latest ones are from 2023. There has been a large body of research on layer-wise and training dynamics topics in the past three years, even in mainstream AI conferences, which are not discussed or compared.

The improvements in all tables are extremely small, mostly within the range of random variance.

Figure 4 claims that “SNVR does not increase training time,” but the paper does not specify the time units, whether the batch size is consistent, or report variance across runs.

The ablation study is incomplete: it does not show the results of using SNVR alone, SNR alone, or their combination.

**Questions:**

Please refer to Weakness

---

### Note · Authors · 2025-11-14

**Comment:**

We sincerely thank all the reviewers for their valuable time and feedback. After careful consideration, we believe that our paper is not yet ready for publication and have decided to withdraw the submission. We will carefully review the suggestions and work to improve our manuscript. Thank you again for your understanding.

**Withdrawal Confirmation:**

I have read and agree with the venue's withdrawal policy on behalf of myself and my co-authors.